# Voltage-Gated Sodium Channel Inhibition by µ-Conotoxins

**DOI:** 10.3390/toxins16010055

**Published:** 2024-01-18

**Authors:** Kirsten L. McMahon, Irina Vetter, Christina I. Schroeder

**Affiliations:** 1Institute for Molecular Biosciences, The University of Queensland, Brisbane, QLD 4072, Australia; 2The School of Pharmacy, The University of Queensland, Woolloongabba, QLD 4102, Australia; 3Genentech, 1 DNA Way, South San Francisco, CA 94080, USA

**Keywords:** µ-conotoxin, peptide, structure–activity relationships, venom peptide, disulfide-rich peptide, voltage-gated sodium channels, subtype selectivity

## Abstract

µ-Conotoxins are small, potent pore-blocker inhibitors of voltage-gated sodium (Na_V_) channels, which have been identified as pharmacological probes and putative leads for analgesic development. A limiting factor in their therapeutic development has been their promiscuity for different Na_V_ channel subtypes, which can lead to undesirable side-effects. This review will focus on four areas of µ-conotoxin research: (1) mapping the interactions of µ-conotoxins with different Na_V_ channel subtypes, (2) µ-conotoxin structure–activity relationship studies, (3) observed species selectivity of µ-conotoxins and (4) the effects of µ-conotoxin disulfide connectivity on activity. Our aim is to provide a clear overview of the current status of µ-conotoxin research.

## 1. Introduction

Marine snails of the genus *Conus* are found in tropical and subtropical waters, including the Indian and Pacific Oceans [1]. Cone snails are preferential hunters of fish, molluscs and worms, and possess a harpoon-like envenomation apparatus rapidly deployed to paralyse prey and deter predators [2,3]. Paralysation and/or death of prey and predators is due to fast-acting bioactive components called conotoxins, and envenomation in humans by some species can be fatal [4]. These conotoxins are generally cysteine-rich peptides, and can modulate a diverse range of ion channels and receptors, including voltage-gated sodium (Na_V_) channels, with, at times, high specificity and potency [5,6].

Na_V_ channels are large transmembrane proteins facilitating influx of sodium ions (Na^+^) across the cell membrane in response to changes in transmembrane voltage. The subsequent rapid influx of Na^+^ into excitable cells, including neurons, cardiac and skeletal muscle leads to the generation of action potentials, which in turn propagates electrical signals throughout the body and facilitates muscle contractions or neurotransmitter release. Humans express nine different subtypes of the pore-forming Na_V_ α-subunit (Na_V_1.1–1.9) that are encoded by the *SCNxA* genes (*x* = 1–5, 8–11), all with different expression patterns and physiological roles (Figure 1) [7,8,9]. Inherited or spontaneous genetic mutations can cause abnormal Na_V_ channel behaviour, which in turn causes conditions known as channelopathies (Figure 1) [10,11]. Compelling evidence implicates several Na_V_ channel subtypes—Na_V_1.1, 1.3, 1.6, 1.7, 1.8 and 1.9, found in periphery sensory neurons—in pain perception [12,13]. This provides support for these channels as targets for analgesic development [14,15]. Conotoxins that can efficiently modulate Na_V_ currents could therefore provide much-needed potential drug leads [16,17].

The ability of neurotoxins to act with high affinity and specificity has contributed to our understanding of the structure and function of Na_V_ channels. µ-Conotoxins, characterised by a conserved cysteine-framework (CC–C–C–CC), potently inhibit Na_V_ channel currents by blocking the pore of the Na_V_ channel, preventing passage of Na^+^ [18,19,20]. Other Na_V_ channel modulators derived from cone snail venom include µO-conotoxins, which mediate channel inhibition through interactions with the voltage-sensor domains [21], δ-conotoxins, which delay Na_V_ channel inactivation via interaction with the domain-IV voltage sensor [22] and ι-conotoxins, which affect the voltage-dependence of channel activation [23]. These conotoxins are beyond the scope of this review and will not be covered here.

To date, 27 µ-conotoxins from 15 different species have been functionally characterised to some extent (Table 1), and show distinct selectivity profiles for Na_V_ channels. For recent reviews on µ-conotoxin discovery and selectivity, see [16,24,25]. One of the limitations of the clinical utility of µ-conotoxins is their promiscuous and relatively unselective nature [17,26], most likely arising from the high sequence homology across the pore of the various Na_V_ channel subtypes. In this review, we evaluate the status of mapping interactions between µ-conotoxins and Na_V_ channels and examine how structure–activity relationship studies have furthered our understanding of the molecular basis of subtype selectivity. We also explore species selectivity and discuss some of the inconsistencies in µ-conotoxin inhibition observed among mammalian Na_V_ channels. Finally, we comment on recent work related to the disulfide connectivity of µ-conotoxins and the consequence of disulfide bonds in Na_V_ channel inhibition. This review aims to provide a clear vision of the current status of µ-conotoxin research.

## 2. Mapping µ-Conotoxin and Na_V_ Channel Interactions

In the absence of detailed crystal and cryo-EM structures, researchers utilised methods including mutagenesis studies and molecular dynamics to investigate the binding of µ-conotoxins to Na_V_ channels [45,46,47]. Following the landmark study reporting the cryo-EM structure of the KIIIA/Na_V_1.2 complex [19], this section explores the agreement between computational methods and experimental methods, and whether insights gained from these can be used for rational design to improve subtype selectivity and/or improve potency.

Initial studies of µ-conotoxin binding to Na_V_ channels assumed that the overall net positive charge of µ-conotoxins facilitated long-range electrostatic interactions, which contributed to the binding to [18], and inhibition of, sodium currents [45]. Mutagenesis studies identified a series of positively charged residues extending from the C-terminal region of GIIIA as contributors to channel interactions [48,49,50]. In particular, a strong interaction between Glu758 (NP_037310.2) and Arg13 of GIIIA localised µ-conotoxin binding to the outer rim of the channel vestibule in rat skeletal muscle [51]. In voltage-clamp experiments using *Xenopus* oocytes expressing rat (r)Na_V_1.2, KIIIA and tetrodotoxin (TTX) were shown to simultaneously bind to neurotoxin binding site 1 and act in concert [52]. Bound KIIIA caused an antagonistic interaction (slowed TTX binding), and a synergistic effect (slowed TTX dissociation), suggesting that TTX bound in the space between KIIIA and the selectivity filter [52]. This co-occupancy was further confirmed with the superimposition of cryo-EM complexes which showed the amine group of Lys7 in KIIIA overlapping with the binding site of the TTX guanidium moiety [19,53]. In contrast to KIIIA, molecular dynamics investigations of GIIIA at rNa_V_1.4 suggest that Arg13 of GIIIA does not reach the selectivity filter, and, consistent with initial mutagenesis studies [54], interacts more strongly with residues in the outer vestibule than the selectivity filter [55]. A follow-up study systematically modelled binding of additional µ-conotoxins, including PIIIA, KIIIA and BuIIIB to rNa_V_1.4, and suggested that a pair of two basic residues (Arg and Lys or Arg and Arg) interact with the charged EEDD ring (Glu403, Glu758, Asp1241 and Asp1532 in rNa_V_1.4; NP_037310.2) within the channel pore [56]. Computational studies using GIIIA and two bacterial channels—Na_V_AB and Na_V_Rh—identified two residues important for interactions, Lys11, which interacts with residues in the selectivity filter of Na_V_Ab, and Arg13, which interacts with two serines of Na_V_Rh [57].

Interactions with human Na_V_ (hNa_V_) channel homologues were evaluated by Chen et al., who developed an hNa_V_1.4 model from the crystal structure of Na_V_Ab and used molecular dynamics simulations to determine Na_V_ channel interactions with PIIIA [58]. Two distinct modes showed either Lys9 or Arg14 of PIIIA predominately occluding the selectivity filter. For the Lys9 example, the side chain of Lys9 was rapidly pulled in towards the filter, whereas other residues interacted favourably with the vestibular wall of the channel. In contrast, when Arg14 orientated towards the filter, four salt bridges formed: one weak interaction inside the filter (Glu180) and three on the vestibule wall [58]. In 2018, a study explored all additional human subtypes aiming to explain subtype selectivity [59]. The prokaryotic Na_V_MS crystal structure [60] was used as a homology structure to generate a model of hNa_V_1.4 and, subsequently, all other subtypes [59]. By studying the interactions between the different subtypes with PIIIA using molecular dynamics simulations, several residues were identified as significantly contributing to binding affinity and were selected as potential influencers of specific TTX-sensitive (TTX-s) (Na_V_1.1–1.4, 1.6 and 1.7) or TTX-resistant (TTX-r) (Na_V_1.5, 1.8 and 1.9) channel inhibition [59]. In particular, the Lys17 side chain of PIIIA appeared to ‘stack’ with the side chains of aromatic channel residues at position 401 (hNa_V_1.4 numbering; either Tyr or Phe) through van der Waals contact [59]. These aromatic residues are absent in TTX-r channels, and suggests the absence of tight interactions resulted in the loss of inhibition observed within TTX-r channels. In TTX-s channels, a neutral Asn residue at position 404 was shown to form energetically favourable hydrogen-bonding to Arg14 of PIIIA [59]. Notably, in complex with the TTX-r channels, which contain a charged Arg or Lys at this position, the hydrogen bonds between PIIIA and the Na_V_ channels were absent [59]. Another residue of interest was Asp1241 (hNa_V_1.4 numbering), which formed favourable electrostatic interactions with Lys17 of PIIIA. This channel residue is conserved in all subtypes except for hNa_V_1.7, where it is replaced by an Ile [59]. Furthermore, a non-conserved neighbouring residue 1536 (hNa_V_1.4) was also shown to be important in some subtypes. For example, in hNa_V_1.4, Asn1536 formed a hydrogen bond with hydroxyproline (Hyp)18 of PIIIA, while at hNa_V_1.6, in the equivalent position, a Lys residue stacked with the µ-conotoxin PIIIA side chain Arg20 [59]. Additionally, despite sharing a Ser residue at this position, hNa_V_1.8 formed pairwise interactions with Arg20 of PIIIA, but did not form the same interactions in the hNa_V_1.9 complex [59]. Together, these findings highlight that both conserved and non-conserved residues may act synergistically to influence the subtype specificity of PIIIA.

Seminal work by Pan et al. solving the KIIIA/hNa_V_1.2 cryo-EM study confirmed several of these predicted interactions discussed above [19]. KIIIA fills the funnel-shaped cavity of the channel pore, interacting closely with domains I, II and III while leaving a small gap near domain IV. In the structure, Lys7 in KIIIA extends into the channel pore to occlude the outer mouth of the selectivity filter by interacting with Glu945, consistent with previous findings [51]. Other main interactions identified between KIIIA and hNa_V_1.2 include Asn3 with Glu330, Lys7 and His12 with Asp949, and Trp8 with Tyr362, as well as Arg14 with Leu920 and Tyr1443. As not all these interactions were consistent with the predicted models, it may suggest that different µ-conotoxins possess different binding modes, compared to KIIIA. Furthermore, Kimball et al. used homology modelling and docking simulations to create a structural model of KIIIA in complex with hNa_V_1.7 [61]. Double-mutant cycle analysis identified specific pairwise toxin–channel interactions, including Lys7 and Glu919, as well as His12 and Asp923, consistent with those established in the KIIIA/hNa_V_1.2 complex [61]. Additionally, the Arg10 and Ile1399 interaction unique to hNa_V_1.7 was proposed to underlie the structural basis for the KIIIA block at hNa_V_1.7 [61].

Comparison of available three-dimensional (3D) µ-conotoxin structures in the M4- and M5-branch of the M-superfamily show that although µ-conotoxins share considerable sequence homology (Table 1), their NMR solution structures display several different and interesting features (Figure 2). The defining feature separating these µ-conotoxin sub-families is the number of residues between the fourth and fifth cysteines. The M4-branch peptides contain four residues, whereas the M5-branch peptides contain five residues [62]. A series of positively charged residues extend from the C-terminal half of µ-conotoxins; on the M5-branch, they appear to form an extended helical shape, whereas the M4-branch displays a truncated helix. The exception is SmIIIA, which appears to generate an α-helix-like structure with a short distance (≥2.4 Å) between Arg16 HN and Trp14 O [63], indicating a possible three-turn helix that is unable to be visualised or predicted in programmes such as MolMol [64] and PyMol. The N-terminal residues of the M4-branch µ-conotoxins appear to elongate their structures. These small, yet significant, differences may influence the variations in potency and subtype selectivity observed between µ-conotoxins. To explore these differences further, researchers have extensively turned to structure–activity studies, which will be discussed in the next section.

## 3. Optimising Potency and Selectivity of µ-Conotoxins by Structure–Activity Relationship Studies

The relatively small size and peptidic nature of µ-conotoxins make them ideal candidates for studying structure–activity relationships that can provide insight into residues that are required for Na_V_ channel inhibition and influence subtype preferences (Figure 3). One approach commonly used is Ala-scanning or Ala-replacement, whereby each non-Cys or Gly residue is systematically replaced with an Ala residue. When investigated through radioligand binding studies, Ala-replacement of Glu15 in TIIIA resulted in a ten-fold increase in inhibition of the skeletal subtype rNa_V_1.4 (pIC_50_ 10.1 ± 0.1 M) and a slight improvement across the neuronal subtype rNa_V_1.2 (pIC_50_ 10.2 ± 0.1 M), suggesting a negative charge at this position created unfavourable interactions with rNa_V_ channel subtypes [30]. Neutral or basic C-terminal extensions, coupled with the Glu15Ala mutation in TIIIA, increased the mutated peptide’s preference for rNa_V_1.2, although overall potency was decreased [65]. In further radioligand binding experiments with SIIIA, Ala-replacement of Lys11, Trp12, Arg14, His16, or Arg18 highlighted the significant role of the C-terminal helix for channel inhibition [40]. Specifically, analogue His16Ala decreased binding affinity for both rat neuronal (rNa_V_1.2) and muscle subtypes (rNa_V_1.4) [40]. In KIIIA, two-electrode voltage-clamp (TEVC) experiments performed on oocytes showed that Ala-replacement of Trp8, Arg10, His12 and Arg14 contributed to rNa_V_1.2 preference [66]. Moreover, Ala-replacement studies of KIIIA performed by whole-cell patch-clamp recordings on HEK293 cells showed that Lys7 (rNa_V_1.2 K_d_ 0.17 ± 0.02 µM > rNa_V_1.4 K_d_ 1.32 ± 0.02 µM > hNa_V_1.7 K_d_ 3.17 ± 0.03 µM) and Arg10 (rNa_V_1.2 K_d_ 0.16 ± 0.03 µM > rNa_V_1.4 K_d_ 1.0 ± 0.02 µM > hNa_V_1.7 K_d_ 1.3 ± 0.03 µM) both lowered K_d_ values and decreased maximal block, but maintained the selectivity profile of native KIIIA (rNa_V_1.2 K_d_ 0.005 ± 0.02 µM > rNa_V_1.4 K_d_ 0.04 ± 0.01 µM > hNa_V_1.7 K_d_ 0.01 ± 0.06 µM) [67]. Interestingly, Ala-replacement of His12 in KIIIA reduced potency and changed the selectivity profile (rNa_V_1.2 K_d_ 10.79 ± 0.02 µM > hNa_V_1.7 K_d_ 19.28 ± 0.02 µM > rNa_V_1.4 K_d_ 110.50 µM), and Ala-replacement of Arg14 completely shifted the selectivity of KIIIA, to favour hNa_V_1.7 (hNa_V_1.7 K_d_ 0.51± 0.04 µM > rNa_V_1.2 K_d_ 1.08 ± 0.04 µM > rNa_V_1.4 K_d_ 5.69 ± 0.02 µM), although with reduced potency [67]. In the case of GIIIA, difficulties with synthesising analogues using Ala-replacement of Hyp residues in GIIIA suggested that these residues also play a role in peptide folding [68]. The synthesis issue was resolved by the use of non-natural amino acid derivatives, including N-methylalanine and sarcosine [68]. The inhibitory effects on twitch contractions of rat diaphragm showed that the hydroxyl group at side chains of Hyp residues were not essential for activity, and instead may play a role in peptide folding [68].

Structure–activity relationships using natural amino acids, in addition to Ala, provided further insights into additional essential residues in KIIIA. In TEVC experiments using oocytes, Trp8 replacement with Leu decreased maximal inhibition of [Trp8Leu]KIIIA at both rNa_V_1.2 and rNa_V_1.4 [66], while Arg, Gln or Glu substitutions increased selectivity for neuronal sodium channel subtypes through reduced potency at rNa_V_1.4 [69]. Sato and colleagues designed a series of mutations in which Arg residues were replaced with Lys in GIIIA and Lys with Arg in GIIIB [70]. By evaluating inhibitory effects on twitch contractions of rat diaphragm, [Arg19Lys]GIIIA mutants showed similar effects to the native GIIIA peptide (IC_50_ 0.10 µM), whereas [Arg13Lys]GIIIA analogues showed drastically reduced inhibition (IC_50_ 30 µM) [70]. These results suggest that, out of the Arg residues, Arg13 is most important for activity, whereas Arg19 has minimal importance in GIIIA inhibition [69]. These findings are in agreement with earlier work that showed an [Arg13Lys]GIIIA analogue inhibiting skeletal muscle subtypes with ten-fold lower potency (IC_50_ 1.12 µM), compared to native GIIIA (IC_50_ 0.10 µM) [71]. Similarly, the presence of a negatively charged amino acid at position 21 (Glu21) may explain why the recently characterised µ-conotoxin BuIIID had no effect on hNa_V_1.4 and hNa_V_1.7 current in whole-cell patch-clamp experiments up to 10 µM, despite sharing high sequence homology with all M-5 branch µ-conotoxins described to date (Table 1) [32]. These results further emphasise the importance of a positively charged residue at this position for µ-conotoxin inhibition of Na_V_ channels.

To investigate the correlation between the size of the Arg-group at position 7 of KIIIA and inhibition efficacy, Walewska and colleagues incorporated non-natural N-substituted Gly monomers (e.g., N-methylglycine (Sar), N-butylglycine and N-octoylglycine) in Lys7 position in KIIIA [72]. In TEVC experiments conducted on oocytes, all analogues negatively affected inhibition kinetics and affinities; however, the [Lys7Sar]KIIIA analogue (K_d_ 0.055 ± 0.03 µM) retained the ability to inhibit rNa_V_1.2, comparable to KIIIA (K_d_ 0.053 ± 0.005 µM) [72,73]. These results agreed with earlier work that identified that a Nle substitution at Lys7 could efficiently mimic the steric bulk of this residue [66].

Structure–activity relationships using KIIIA helped identify residues located on the α-helical region that are critical for Na_V_1.2 and Na_V_1.4 inhibition [66,74], and several studies have thus focused on the conserved residues residing in the C-terminal half of µ-conotoxins. Stevens and colleagues designed small peptides (12–16 amino acids), based on the predicted pharmacophore of KIIIA, which did not lose potency and selectivity [75]. The most promising compound was a disulfide-deficient (CysI/CysIV) analogue, which improved potency at rNa_V_1.2 (IC_50_ 34.1 ± 0.01 nM) [75]. Unfortunately, efforts to isolate key residues from the pharmacophore region were not as successful on other channel subtypes. Optimised three-residue minimised KIIIA peptidomimetic derivatives did not improve the potency of KIIIA at rNa_V_1.7 [76]. The best peptidomimetic only produced a 20% block of rNa_V_1.7 current at 100 µM [76].

In contrast to the highly conserved µ-conotoxin C-terminal, the N-terminal half of µ-conotoxins is much less conserved, particularly within the M5-branch of µ-conotoxins (Table 1). This region is also more flexible than the C-terminal portion, as demonstrated by NMR relaxation experiments on µ-conotoxin SIIIA [77]. Thus, µ-conotoxins appear to be more amenable to modifications within the N-terminal region. In GIIIA, incorporation of a MeBzl-protected Cys residue at Tyr5 did not affect potency as much as Arg13 modifications, and suggests a position within µ-conotoxins that may be adaptable to the incorporation of a fluorescent tag for visualisation within cells [78]. Replacement of residues in loop one (between Cys2 and Cys3) in SIIIA with non-natural backbone spacers produced longer-lasting analgesic effects in a murine model of inflammatory pain than native SIIIA [79]. In the case of SxIIIC, analogues with a loop-one truncation ([∆7,8]SxIIIC) reduced potency at hNa_V_1.7 (IC_50_ 313.6 ± 36.6 nM), compared to native SxIIIC (IC_50_ 152.2 ± 26.8 nM) [17,44]. Interestingly, under identical electrophysiology assay conditions at hNa_V_1.7, these IC_50_ values were similar to KIIIA (IC_50_ 383.5 ± 15.2 nM) [44]. As KIIIA has a similarly truncated loop 1, it was proposed that the additional loop-one residues in SxIIIC are required for increased inhibition of hNa_V_1.7, when compared to KIIIA [44].

µ-Conotoxin N-terminal extensions (residues prior to Cys1) have also been evaluated for their contribution to potent inhibition and subtype selectivity. In radioligand binding assays, subtype discrimination of SIIIA for rNa_V_1.2 over rNa_V_1.4 was achieved through the removal of the N-terminal pyroglutamic (Pyr) residue [40]. Furthermore, a triple SIIIA mutant with truncated N-terminal, Asn5Lys and Asp15Ala substitutions improved inhibition ~20-fold (pIC_50_ 9.2 ± 0.09 M) at rNa_V_1.2, compared to native SIIIA (pIC_50_ 8.02 ± 0.12 M) [40,80]. In the case of BuIIIB, in oocytes expressing rNa_V_1.3 the removal of the N-terminal extension resulted in a four-fold improvement in inhibition (K_d_ 0.053 ± 0.013 µM) [81]. Ala-replacement of Gly2 and Glu3 in BuIIIB also improved potency at rNa_V_1.3 (K_d_ 0.036 ± 0.009 µM and 0.015 ± 0.003 µM, respectively) [81,82], and the use of D-Ala at position 2 resulted in 40-fold higher potency (K_d_ 0.036 ± 0.009 µM) at rNa_V_1.3, compared to native BuIIIB (K_d_ 0.2 ± 0.086 µM) [26,81]. However, when explored in the pharmacologically favourable subtype hNa_V_1.7, truncated SxIIIC did not significantly affect potency (IC_50_ 106.2 ± 19.4 nM), compared to native SxIIIC [44].

µ-Conotoxins from *C. geographus,* GIIIA, GIIIB and GIIIC, were among the first to be identified [27]. Recent work evaluating the structure–activity relationships of these natural analogues have identified key contributors to Na_V_ channel potency and selectivity. The NMR solution structure and whole-cell patch-clamp electrophysiology studies of GIIIC highlight the role of non-pore interacting residues in influencing Na_V_ channel potency [41]. GIIIC differs by only a few residues from GIIIA and GIIIB, yet GIIIC (IC_50_ 286 ± 13 nM) has a higher IC_50_ than GIIIA (IC_50_ 110 ± 4 nM) at hNa_V_1.4. GIIIC showed minimal affinity towards mouse (m)Na_V_1.6 (~14% inhibition at 1 µM), and was not effective against rNa_V_1.2, hNa_V_1.5, hNa_V_1.7 and hNa_V_1.8, up to 1 µM [41]. Differences in potency are likely due to the structural differences in the N-terminal region, as critical charged residues between GIIIA and GIIIC are conserved. At position 18, a substitution of an uncharged Gln in GIIIA to a hydrophobic Met in GIIIB, or Leu in GIIIC, contributes to significant structural differences. Leu18 appears to orientate towards the core of GIIIC and, through increased tethering to the N-terminal reducing solvent accessibility, possibly contributes to the decrease in potency observed for GIIIC at rNa_V_1.4 [41]. These findings demonstrate the additional structural space to explore when investigating subtype selectivity.

Together, these examples show how structure–activity relationships have identified residues involved in potent and selective Na_V_ channel inhibition. However, the lack of consistency among studies, particularly regarding Na_V_ species, subtypes and assay conditions, make it challenging to ascertain the specific regions of interest of the peptides and the full therapeutic potential of µ-conotoxins.

**Figure 3 toxins-16-00055-f003:**
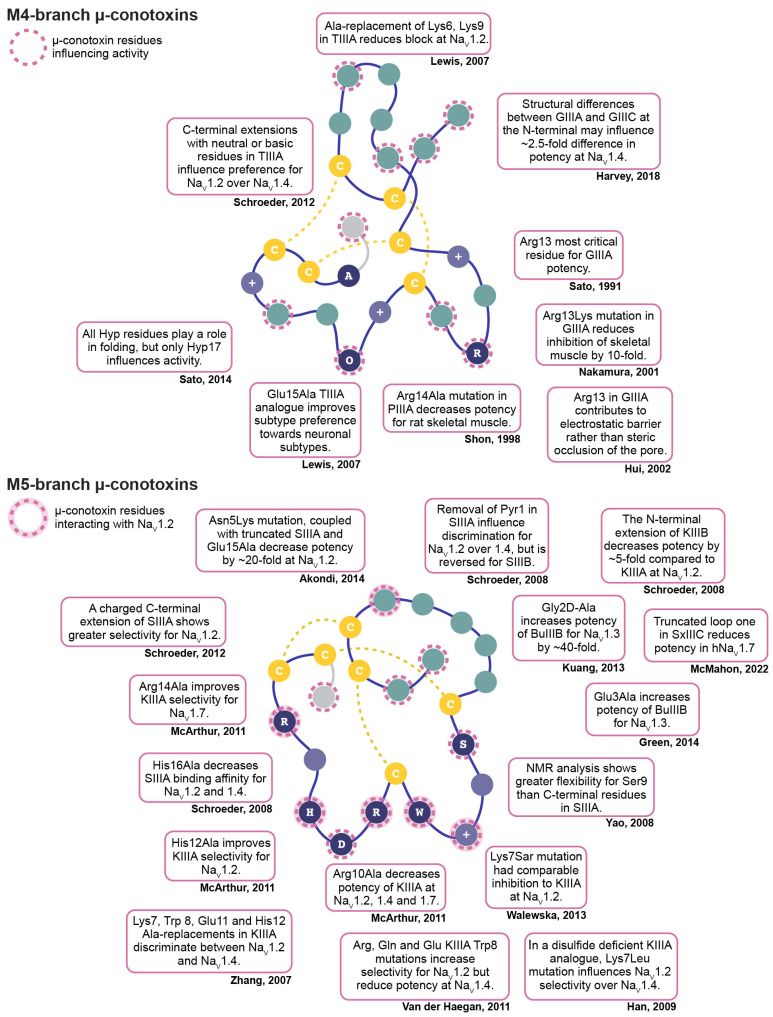
Summary of µ-conotoxin residues influencing potency and selectivity, as determined by structure–activity relationship studies. Representative globular fold of M4-branch (**top** panel) and M5-branch (**bottom** panel) µ-conotoxins, as modelled by available NMR structures. Conserved residues (>90%) are represented by single amino acid letter (navy circles) or + (either Arg or Lys; purple circles). Non-conserved residues are not annotated (teal circles). Cysteine residues (yellow circles) and disulfide bonds (yellow dashed line) are represented with canonical connectivity. Structure–activity relationship studies have identified several residues that modulate inhibition (pink dashed rings). Summary of the key findings are shown in neighbouring boxes. Hyp—hydroxyproline, Pyr—pyroglutamate and Sar—N-methylglycine. For clarity, disulfide bond mutations, which will be discussed in the following section, are not included. (Hui, 2002 [18]; Shon, 1998 [28]; Lewis, 2007 [30]; Schroeder, 2008 [40]; Harvey, 2018 [41]; McMahon, 2022 [44]; Sato, 1991 [48]; Schroeder, 2012 [65]; Zhang, 2007 [66]; MaArthur, 2011 [67]; Sato, 2014 [68]; Van der Haegan, 2011 [69]; Nakamura, 2001 [71]; Walewska, 2013 [72]; Yao, 2008 [77]; Akondi, 2014 [80]; Kuang, 2013 [81]; Green, 2014 [82]; Han, 2009 [83]).

## 4. Species Specificity of µ-Conotoxin Inhibition

One challenging aspect of translating µ-conotoxins into analgesics is the, at times, striking difference in µ-conotoxin potency at Na_V_ channels observed between species, which was first observed for GIIIA interacting with rat and human Na_V_ homologues. Using electrophysiological recordings of *Xenopus* oocytes expressing human skeletal muscle Na^+^ channels (hSkM1), GIIIA potency was 22-fold lower than in the rat homologue, rSkM1 [84,85]. Furthermore, the initial characterisation of KIIIA, SIIIA and SmIIIA showed that these µ-conotoxins were effective against TTX-r DRG in frog neurons [36,38]. However, this observation did not extend to mouse and rat TTX-r isoforms, where these µ-conotoxins were active only at selected TTX-s Na_V_ channels [26,40,66]. A recent study reporting BuIIIB activity at hNa_V_1.1–1.8 using whole-cell patch-clamp electrophysiology [32] showed that the selectivity profile of 1.3 ≈ 1.4 < 1.7 ≈ 1.1 ≈ 1.2 ≈ 1.6 varied from that previously determined for rNa_V_ channels using TEVC experiments (1.4 < 1.2 < 1.3 < 1.1 < 1.6 < 1.5) [26]. While differences between techniques could possibly explain the differences in Na_V_ channel activity, it is likely that µ-conotoxin binding is different between homologous channels of different species.

As sequence differences are minimal among mammalian homologues (Figure 4), the species selectivity of some µ-conotoxins is likely to arise from a few key substitutions. In a study by Cummins et al., different kinetic properties of GIIIA and GIIIB binding were suggested to contribute to differential effects on Na_V_1.4 from different species [47]. A 20-fold difference in GIIIA potency at rNa_V_1.4 (IC_50_ 58 ± 5 nM) and hNa_V_1.4 (IC_50_ 1228 ± 139 nM) was found to be influenced by a mutation (Ser to Lys) in the domain II pore-loop at position 729 of rNa_V_1.4 [47]. Furthermore, at position 732 of rNa_V_1.4, an Asp-to-Lys mutation reduced potency 20-fold for GIIIB (IC_50_ 972 ± 137 nM; WT rNa_V_1.4 IC_50_ 49 nM), but only four-fold for GIIIA (IC_50_ 228 ± 37 nM) [47]. These results support a model whereby specific residues on the descending segment of the domain II pore-loop are critical for µ-conotoxin selectivity for rNa_V_1.4.

Species differences within the domain II pore-loops may also contribute to differences in activity of SIIIA at rat and human Na_V_1.7. While SIIIA was reported to display intermediate potency at rNa_V_1.7 [77], it was inactive at hNa_V_1.7 [44]. Leipold and colleagues identified Ala728 as a key residue in the SIIIA/rNa_V_1.4 interaction [86]. Using channel-mutant analysis, the corresponding residue in hNa_V_1.7, Asp889, was shown to contribute to the loss of activity of SIIIA at hNa_V_1.7 [86].

Although not discussed in detail within this review, µ-conotoxins also display differential activity at bacterial Na_V_ channels. For example, Na_V_ channel inhibition by PIIIA varies more than six orders of magnitude between bacterial (sub-picomolar) and mammalian (micromolar) Na_V_ channels [87]. Molecular dynamics studies of PIIIA and the bacterial Na_V_ channel identified multiple binding modes [88]. Chen et al. proposed that the symmetry of the basic residues within PIIIA allowed the µ-conotoxin to bind in various orientations within the bacterial Na_V_ channel, resulting in several different residues blocking the selectivity filter [88].

Altogether, it is clear that differences between species need to be considered during the translation of µ-conotoxins into therapeutics.

## 5. Effects of Disulfide Connectivity

One defining feature of µ-conotoxins is the characteristic type III cysteine framework (CC–C–C–CC) [89]. Early studies using GIIIA utilised BB-dideuterio protected Cys residues and mass spectrometry methods to determine the disulfide connectivity as 1/4, 2/5 and 3/6 for the synthetically produced peptide [90]. As this co-eluted with the native venom, this conformation was canonically described as the ‘native’ conformation for µ-conotoxins [90]. Additional methods to determine connectivity include inter-cysteine NOE analysis, which was used to evaluate SxIIIA disulfide connectivity [29], while direct mass spectrometric collision-induced dissociation fragmentation was used for BuIIIB [81]. When produced synthetically, both SxIIIA and BuIIIB preferentially adopt the 1/4, 2/5 and 3/6 cystine conformation. However, as µ-conotoxin research has expanded, the true ‘native’ conformation has been questioned. Indeed, with six Cys residues, a total of 15 disulfide conformations could occur, and recent work has shed light on the impact these disulfide isomers have on µ-conotoxin activity at Na_V_ subtypes [43,91,92,93,94]. A classic example is KIIIA, which was originally identified from cDNA, and thus no venom extract could be used to confirm native connectivity [36]. Given its sequence similarity to other known µ-conotoxins, molecular modelling of SmIIIA suggested that KIIIA was likely to have the same connectivity [36,66]. However, in later studies, Khoo et al. utilised PADLOC [95], a method detecting patterns of disulfides from local NMR constrains, to identify that KIIIA displayed two alternate disulfide connectivities, which were ultimately resolved to be a major isomer with connectivity of 1/5, 2/4 and 3/6, in addition to a minor isomer with connectivity 1/6, 2/4 and 3/5 [37]. Surprisingly, neither of the two isomers display the ‘native’ disulfide bond framework. These different KIIIA isomers have subsequently been shown to possess different potencies at a number of hNa_V_ channel subtypes [43]. Likewise, BIIIA with cystine connectivity 1/5, 2/4 and 3/6 showed reduced potency at hNa_V_1.2 and increased potency at hNa_V_1.4, compared to BIIIA, with canonical cysteine connectivity (1/4, 2/5 and 3/6) in whole-cell patch-clamp experiments [32]. The data indicate that some μ-conotoxins can not only tolerate different disulfide connectivity, but that this can lead to subtle but important changes in their selectivity profiles.

In attempts to delineate the impact of different disulfide connectivity on µ-conotoxin activity, researchers have generated disulfide-deficient µ-conotoxin KIIIA analogues, where they systematically removed each of the disulfide bonds and examined the structure and activity of the peptides [74,83]. Using TEVC experiments in oocytes, Khoo et al. demonstrated that KIIIA lacking the first disulfide bond, KIIIA[C1A,C9A] retained activity at rNa_V_1.2 (KIIIA K_d_ 0.005 ± 0.005 µM and KIIIA[C1A,C9A] K_d_ 0.008 ± 0.002 µM) [74]. Similarly, Han et al. showed that deletion of the first bridge of KIIIA retained the ability to block rNa_V_1.2 (KIIIA 90 ± 3% and μ-KIIIA[C1A,C9A] 93 ± 2%) and rNa_V_1.4 (KIIIA 86 ± 6% and μ-KIIIA[C1A,C9A] 85± 2%) [83]. Furthermore, with the removal of the N-terminal residue and the inclusion of a backbone spacer, the resulting minimised analogue [desC1]KIIIA[S3/4Aopn,C9A] produced analgesia in murine inflammatory models [95]. These findings lay the groundwork for the design of minimised µ-conotoxin peptides and the development of hybrid BuIIIC-KIIIA analogues, which were designed based on the premise that the first KIIIA disulfide could be deleted without losing potency and selectivity [75]. A series of miniaturized analogues led to the development of R2-Midi, a 14-residue peptide with two disulfide bonds that yielded an IC_50_ 34 ± 0.01 nM in TEVC in oocytes expressing the rNa_V_1.2 isoform [75]. Notably, these experiments were conducted prior to the identification of KIIIA disulfide isomers and the effects of disulfide deletion. Recently, in whole-cell patch-clamp experiments at hNa_V_1.7, conducted by Zhao et al., a disulfide-deficient KIIIA analogue [C1A, N3Dab, C15R]KIIIA, which lacked the 1/5 disulfide bond, was shown to be approximately four-fold more potent (IC_50_ 110.75 ± 20.42 nM) than the KIIIA isomer I (1/5, 2/4, 3/6; IC_50_ 413 ± 71 nM) and eight-fold more potent than the ‘native’ KIIIA (1/4, 2/5, 3/6) [96]. In a study exploring the effects of disulfide deletions of PIIIA using electrophysiology experiments at hNa_V_1.4, all three analogues significantly lost activity when each of the ‘native’ disulfide bonds were removed [93]. These results suggest that, unlike other µ-conotoxins, PIIIA needs all three disulfide bonds in the ‘native’ conformation, to retain activity [93].

The role of disulfide connectivity has also been evaluated for structural stability. Sato and colleagues systematically replaced Cys pairs with Ala in GIIIA, and identified loss of activity in a rat diaphragm twitch-contraction assay [68]. Observed shifts in circular dichroism spectra suggest that all three bonds are needed to stabilise the active conformation of GIIIA [68]. When assessed at rNa_V_1.4, using electrophysiology assays, the disulfide bond 3/6 significantly influenced the rigidity and resulting activity of GIIIA [97]. Investigations with PIIIA also confirmed the importance of the 3/6 bond for hNa_V_1.4 inhibition [94].

In work by Heimer et al., all possible 15 isomers of PIIIA were synthesised and characterised structurally, revealing major changes in both global conformation and backbone flexibilities [91]. Disulfide bond patterns typical of conotoxins (e.g., 1/4, 2/5, 3/6) resulted in more compact folds. In contrast, isomers which connected the N-terminal cysteines (1/2) or C-terminal cysteines (4/5) resulted in highly flexible structures. A follow-up study evaluated the activity of these analogues at hNa_V_1.4 and found that while some isomers retained activity, none were more potent than the ‘native’ conformation (1/4, 2/5, 3/6; IC_50_ 105.3 ± 29.9 nM) [93]. Interestingly, these findings are in contrast to earlier studies looking at PIIIA isomers at rNa_V_1.4, which found that the minor disulfide isomer (1/5, 2/6, 3/4) increased potency (IC_50_ 46.7 ± 6.5 nM) [98]. Structural differences evaluated using NMR and molecular dynamics indicated that the conformational isomers left large portions of the hNa_V_1.4 channel pore exposed, as compared to the native PIIIA, which covered the central pore [93]. Given the aforementioned trend of Na_V_ channel activity differing between mammalian species, it is likely that the favoured µ-conotoxin conformation isomer also contributes to the differences observed between species. A complex study combining µ-conotoxin disulfide isomers and different mammalian Na_V_ channels could provide interesting insights into these differences.

Another study of interest looked at classifying the folding pathways of µ-conotoxins [92]. Using NMR structures and applying molecular dynamics simulations to break and re-form the disulfide bonds, Paul George et al. monitored the stability of µ-conotoxins during peptide refolding [92]. µ-Conotoxins GIIIA and SxIIIC showed a single oxidation peak, indicating a preference for folding back into a single conformation [92]. On the other hand, SmIIIA and PIIIA appeared to favour two or more conformations, and showed a slower folding rearrangement [92]. Thus, noncovalent interactions and electrostatic forces may contribute to these slower or more challenging folding arrangements [91]. Interestingly, this report also observed one oxidation peak for KIIIA, which is in contrast to reports of KIIIA producing two oxidation peaks [37,43]. Differences in batch size and composition of the oxidation buffer may contribute to this effect.

Collectively, these studies highlight the importance of structural studies and disulfide bond mapping to correctly identify µ-conotoxin isomers during synthesis for accurate evaluation of µ-conotoxin activity. Continued work in this area may support the large-scale production of µ-conotoxins for therapeutic use.

## 6. Conclusions and Future Directions

µ-Conotoxins represent a rich source of potential new drug-leads for the treatment of Na_V_ channel channelopathies, including pain. Not only do µ-conotoxins make promising candidates for therapeutics, but they also provide excellent tools for studying Na_V_ channel structure and function. However, additional studies addressing the molecular basis of species selectivity and differences in the activity of disulfide isomers, are required. Additional considerations not discussed in this review include the co-expression of various Na_V_ β-subunits with the pore-forming α-subunit and promiscuous off-target (i.e., non-Na_V_ channel) activity, as this may lead to undesirable side-effects. The inconsistency in co-expression of β-subunits among studies makes it challenging to directly compare different bodies of work, as the co-expression of β-subunits can significantly affect affinity and efficacy of µ-conotoxins [99]. Furthermore, µ-conotoxins have been shown experimentally to inhibit other ion channels, including neuronal nicotinic acetyl choline receptors (CnIIIC) and voltage-gated potassium channels (PIIIA) [35,100]. However, given the high concentrations (10 µM PIIIA) used in the latter study, this may be less concerning when assessing the therapeutic potential of µ-conotoxins, which generally inhibit Na_V_ channel response at nM levels.

While bacterial and mammalian homologues, including human, rat and mouse, have contributed to, and will undoubtedly continue to add to, our understanding of the mechanisms behind Na_V_ channel blockage, inconsistencies between species highlight the need to specifically evaluate activity in human homologues for the development of µ-conotoxin-derived therapeutics. Furthermore, given the activity and structural differences between the two branches of µ-conotoxins mentioned here, we recommend these two-branches of µ-conotoxins be evaluated independently of each other.

Considering that over 750+ *Conus* species have been identified to date, including recent work extracting new µ-conotoxins sequences from historical specimens, and that only a small handful of µ-conotoxins have been functionally characterised, it is likely that a large number of µ-conotoxins remain unexplored [32,101]. Given the advances in structure–function technology, particularly landmark structural studies which provide new insights into channel blockage mechanisms in humans, the field of µ-conotoxin research has much more to offer and offers great potential to develop subtype selective Na_V_ channel inhibitors.

## Figures and Tables

**Figure 1 toxins-16-00055-f001:**
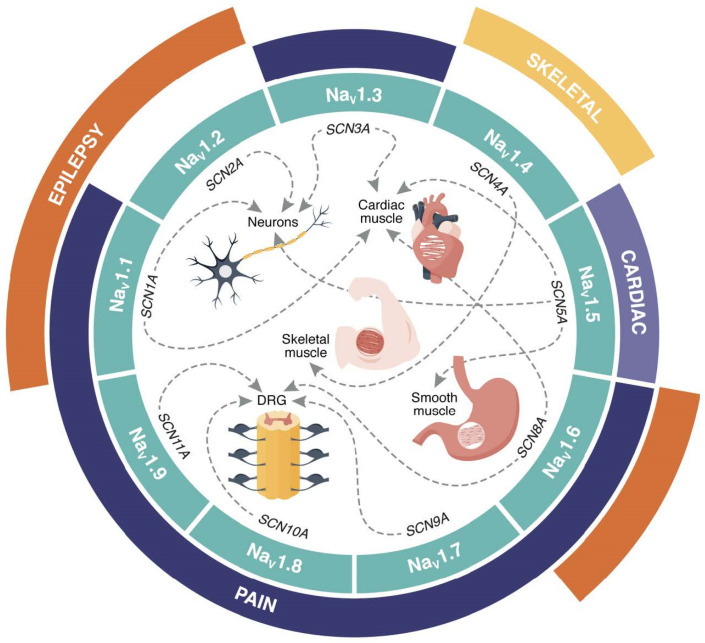
Distribution of human Na_V_ channels and associated channelopathies. The nine Na_V_ channel subtypes (Na_V_1.1–1.9; teal segments) are encoded by a corresponding *SCNxA* gene (where *x* = 1–5, 8–11). Major tissues expressing various Na_V_ subtypes are indicated by dotted lines. Outer segments denote channelopathies associated with Na_V_ channel mutations (pain—dark blue, cardiac—purple, epilepsy—orange, and skeletal muscle disorders—yellow). DRG—dorsal root ganglia.

**Figure 2 toxins-16-00055-f002:**
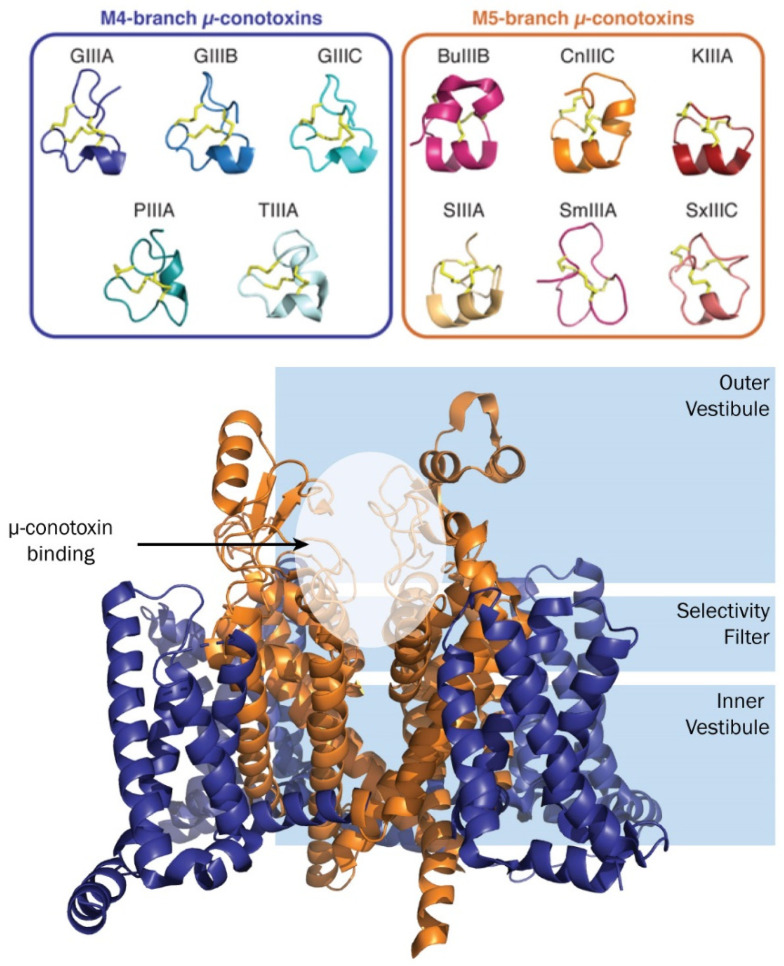
Representations of µ-conotoxin 3D NMR structures determined to date (**top**) and illustrative µ-conotoxin–Na_V_ channel binding site (**bottom**). Coordinates from the PDB (1TCJ GIIIA; 1GIB GIIIB; 6MJD GIIIC; 1R9I PIIIA; 2LOC BuIIIB, 2YEN CnIIIC; 2LXG KIIIA; 1Q2J SmIIIA, 6X8R SxIIIC and 6J8J hNa_V_1.7) and BMRB (20024 TIIIA and 20025 SIIIA) were used to generate figures in PyMol.

**Figure 4 toxins-16-00055-f004:**
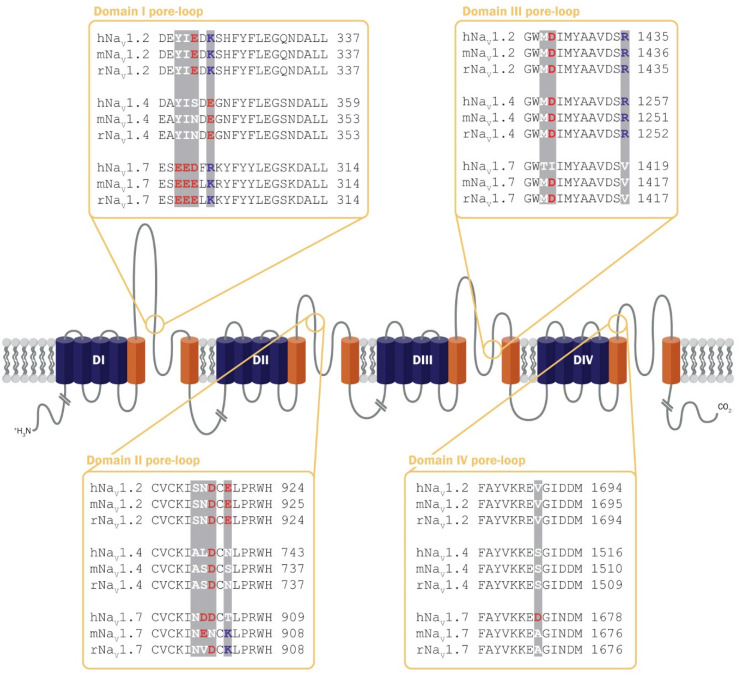
Sequence differences within the µ-conotoxin Na_V_ channel binding site may confer species specificity. Only minimal sequence differences are observed between mammalian species within the Na_V_ channel binding site. For clarity, only the human, rat, and mouse sequences for three subtypes are shown, including the frequently studied Na_V_1.2 and Na_V_1.4, as well as the therapeutically relevant Na_V_1.7. Sequence differences are highlighted by shaded bars. Residues are coloured according to properties (positively charged—blue; negatively charged—red and neutral—white).

**Table 1 toxins-16-00055-t001:** Summary of µ-conotoxins functionally characterised to date.

µ-Conotoxin	Species	Sequence	Activity ^#^	PDB/BMRB	Discovery
**M4-branch**					
GIIIA	*Conus geographus*	RD**CC**TOOKK**C**KDRQ**C**KOQ**CC**A *	rNa_V_1.4 ^a^	ITCJ	[27]
GIIIB	*Conus geographus*	RD**CC**TOORK**C**KDRR**C**KOMK**CC**A*	rNa_V_1.4	1GIB	[27]
GIIIC	*Conus geographus*	RD**CC**TOOKK**C**KDRR**C**KOLK**CC**A *	rNa_V_1.4 ^b^	6MJD	[27]
PIIIA	*Conus purpurascens*	ZRL**CC**GFOKS**C**RSRQ**C**KOHR**CC**A *	rNa_V_1.4 ^a^	1R9I	[28]
SxIIIA	*Conus striolatus*	R**CC**TGKGS**C**SGRA**C**KNLK**CC**A *	rNa_V_1.4	-	[29]
SxIIIB	*Conus striolatus*	ZK**CC**TGKGS**C**SGRA**C**KNLK**CC**A *	rNa_V_1.4	-	[29]
TIIIA	*Conus tulipa*	RHG**CC**KGOKG**C**SSRE**C**ROQH**CC** *	rNa_V_1.4	20024	[30]
TsIIIA	*Conus tessulatus*	G**CC**RWP**C**PSR**C**GMAR**CC**SS	hNa_V_1.8 ^c^	-	[31]
**M5-branch**					
AdIIIA	*Conus adamsonii*	RPV**CC**TGKGKRG**C**SSRW**C**RDHSR**CC** *	hNa_V_1.4	-	[32]
BIIIA	*Conus bruuni*	**CC**NKNGG**C**AGKW**C**KGRSR**CC** *	hNa_V_1.2	-	[32]
BuIIIA	*Conus bullatus*	VTDR**CC**KGKRE**C**GRW**C**RDHSR**CC** *	rNa_V_1.4	-	[33]
BuIIIB	*Conus bullatus*	VGER**CC**KNGKRG**C**GRW**C**RDHSR**CC** *	hNa_V_1.3 ≈ 1.4 ^d^	2LOC	[33]
BuIIIC	*Conus bullatus*	IVDR**CC**NKGNGKRG**C**SRW**C**RDHSR**CC** *	rNa_V_1.4	-	[33]
BuIIID	*Conus bullatus*	VGLY**CC**RPKPNGQMM**C**DRW**C**EKNSR**CC** *	N/A	-	[32]
BuIIIE	*Conus bullatus*	VGEH**CC**RPRLRPKPLWRGKRE**C**DRW**C**KSHSR**CC** *	hNa_V_1.3	-	[32]
CIIIA	*Conus catus*	GR**CC**EGPNG**C**SSRW**C**KDHAR**CC** *	Amphibian	-	[34]
CnIIIA	*Conus consors*	GR**CC**DVPNA**C**SGRW**C**RDHAQ**CC** *	rNa_V_1.2 ≈ 1.4	-	[34]
CnIIIB	*Conus consors*	ZG**CC**GEPNL**C**FTRW**C**RNNAR**CC**RQQ	Amphibian	-	[34]
CnIIIC	*Conus consors*	G**CC**NGPKG**C**SSKW**C**RDHAR**CC** *	hNa_V_1.4	2YEN	[35]
DIIIA	*Conus dusaveli*	RKV**CC**DGPNG**C**SSKW**C**KDHAR**CC** *	hNa_V_1.3	-	[32]
KIIIA	*Conus kinoshitai*	**CC**N**C**SSKW**C**RDHSR**CC** *	hNa_V_1.4 ^e^	2LXG	[36]
KIIIB	*Conus kinoshitai*	NG**CC**N**C**SSKW**C**RDHSR**CC** *	rNa_V_1.4	-	[37]
MIIIA	*Conus magus*	ZG**CC**NVPNG**C**SGRW**C**RDHAQ**CC** *	Amphibian	-	[34]
SmIIIA	*Conus stercusmuscarum*	ZR**CC**NGRRG**C**SSRW**C**RDHSR**CC** *	hNa_V_1.4 ^f^	1Q2J	[38]
SIIIA	*Conus striatus*	ZN**CC**NGG**C**SSKW**C**RDHAR**CC** *	rNa_V_1.4	20025	[36,39]
SIIIB	*Conus striatus*	ZN**CC**NGG**C**SSKW**C**KGHAR**CC** *	rNa_V_1.2 ≈ 1.4	-	[40]
SxIIIC	*Conus striolatus*	RG**CC**NGRGG**C**SSRW**C**RDHAR**CC** *	hNa_V_1.4	6X8R	[17]

* C-terminal amidation; ^#^ The Na_V_ subtype at which the individual peptide is the most active. ^a^ [26] ^b^ [41] ^c^ [42] ^d^ [32] ^e^ [43] ^f^ [44] represent selectivity data reported independent of discovery.

## Data Availability

All data used for analysis in this review article are included in the article.

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
