# Peer review of "Voltage-Gated Sodium Channel Inhibition by µ-Conotoxins"

_toxins, 2024, doi:10.3390/toxins16010055_

Round 1
Reviewer 1 Report
Comments and Suggestions for Authors
In the manuscript, the authors reviewed the progress on studying the relationship between sodium channels and conotoxins in recent years. The binding modes of µ-conotoxin with sodium channel, structure-activity relationship study of µ-conotoxins, their species selectivity and disulfide bonding modes were described accordingly, which provide the readers with insight on the future studying the µ-conotoxins for therapeutic development. Therefore, I propose to publish it in toxins after some modifications.
Specific comments:
1 There are several review papers already reported on similar topic. So, authors are required to clearly mention the specificity of this manuscript in comparison with the others.
3 Is Glu758 specifically at one Nav subtype, or at all sodium channels? It is a bit of confusing, although the author gave the name as NP_037310.2. Please clear this (line 80).
4 In the part of "Mapping µ-conotoxin and NaV channel interactions", the authors describe in detail the interactions between KIIIA, PIIIA, etc. and sodium channels. I suggest add a diagram of their binding modes in Fig. 2 to facilitate the readers' understanding.
5 The 15 position of SIIIA is Asp, not Glu (shown in Table 1). In addition, please re-estimate the activity change for the SIIIA mutant in comparison with the native peptide (line 253).
6 Glu carries a charge, and please revise it (line 272).
7 At the end of "Optimising potency and selectivity of µ-conotoxins by structure-activity relationship studies", I also suggested that the listed µ-conotoxins should be compared and summarized to get some of their commonalities, which are also the significance of this review.
8 Figure 4 shows an indication error, and there should be a Domin II pore-loop between DII and DIII, please revise it. Furthermore, please revise the description of the caption 4.
9 KIIIA-1 already represents [C1A,C15A]KIIIA, and therefore it should be [N3Dab,A15R]KIIIA-1 or [C1A,N3Dab,C15R]KIIIA (line 390).
10 References 15, 38, 60 have formatting problems, please correct them.
Comments on the Quality of English LanguageThere are several places where English expression or writing should be revised. Please go through the whole manuscript carefully, and revise the corresponding errors.
Author Response
Reviewer 1
In the manuscript, the authors reviewed the progress on studying the relationship between sodium channels and conotoxins in recent years. The binding modes of μ-conotoxin with sodium channel, structure-activity relationship study of μ-conotoxins, their species selectivity and disulfide bonding modes were described accordingly, which provide the readers with insight on the future studying the μ-conotoxins for therapeutic development. Therefore, I propose to publish it in toxins after some
modifications.
Specific comments:
1) There are several review papers already reported on similar topic. So, authors are required to clearly mention the specificity of this manuscript in comparison with the others.
Response: We thank the reviewer for their review of the manuscript. In contrast to recent NaV channel/μ-conotoxin reviews that focus on a specific topic, this manuscript incorporates commentary from multiple aspects of μ-conotoxin research including recent structural-activity relationship studies and the effects of µ-conotoxin disulfide connectivity on activity, all whilst incorporating the latest discoveries and thus providing a current status as mentioned in lines 56–62.
In this review we evaluate the status of mapping interactions between µ-conotoxins and NaV channels and examine how structure-activity relationships studies have furthered our understanding of the molecular basis of subtype selectivity. We also explore species selectivity and discuss some of the inconsistencies in µ-conotoxin inhibition observed between mammalian NaV channels. Finally, we comment on recent work related to the disulfide connectivity of µ-conotoxins and the consequence of disulfide bonds in NaV channel inhibition. This review aims to provide a clear vision of the current status of µ-conotoxin research.
3) Is Glu758 specifically at one Nav subtype, or at all sodium channels? It is a bit of confusing, although the author gave the name as NP_037310.2. Please clear this (line 80).
Response: The reference publication predates defining of the nine different Nav channel subtypes. To avoid confusion or misleading claims no specific channel has been included. The original publication references a skeletal muscle Na+ channel which corresponds to the protein reference code.
4) In the part of "Mapping μ-conotoxin and NaV channel interactions", the authors describe in detail the interactions between KIIIA, PIIIA, etc. and sodium channels. I suggest add a diagram of their binding modes in Fig. 2 to facilitate the readers' understanding.
Response: We thank the reviewer for their suggestion. We feel to individually list each µ-conotoxin and its binding mode to each of the nine sodium channel subtype would be too complex to be visualised in a static graphic. Furthermore, the variation in binding modes is so subtle they would not be clearly captured at this level and would be more suited to a video or animated format which is beyond the scope of this review.
To aid readers through this section, we have updated Figure 2 to include an illustrative figure of a Nav channel and the approximate location of µ-conotoxin binding. This figure also includes visual representation of the terminology (e.g. outer vestibule, selectivity filter etc.) used throughout this section.
5) The 15 position of SIIIA is Asp, not Glu (shown in Table 1). In addition, please re-estimate the activity change for the SIIIA mutant in comparison with the native peptide (line 253).
Response: Table 1 lists the correct sequence for SIIIA. Line 253 has been corrected Glu15Ala ïƒ Asp15Ala.
6) Glu carries a charge, and please revise it (line 272).
Response: Glu should have read Gln. Line 272 has been updated.
7) At the end of "Optimising potency and selectivity of μ-conotoxins by structure-activity relationship studies", I also suggested that the listed μ-conotoxins should be compared and summarized to get some of their commonalities, which are also the significance of this review.
Response: We thank the reviewer for their suggestion. We feel that to provide a textual summary would be repetitious and it could be misleading to directly compare the different studies as there are a number of factors are associated with evaluating selectivity (for example, the techniques used, species of Nav channels used, preparation of μ-conotoxin or disulfide connectivity of μ-conotoxins). Furthermore, Figure 3 provides a high-level summary of the key results discussed in this section.
8) Figure 4 shows an indication error, and there should be a Domin II poreloop between DII and DIII, please revise it. Furthermore, please revise the description of the caption 4.
Response: Figure 4 updated to fix error.
9) KIIIA-1 already represents [C1A,C15A]KIIIA, and therefore it should be [N3Dab,A15R]KIIIA-1 or [C1A,N3Dab,C15R]KIIIA (line 390).
Response: Updated to KIIIA to keep consistent with other abbreviations.
10) References 15, 38, 60 have formatting problems, please correct them.
Response: Updated references.
There are several places where English expression or writing should be revised. Please go through the whole manuscript carefully, and revise the corresponding errors.
Response: The manuscript has been carefully proofread by a native English speaker.
Reviewer 2 Report
Comments and Suggestions for Authors
This review delivers exactly what it promises. It adeptly addresses a crucial aspect of μ-Conotoxin research, particularly their potential as pharmacological probes and analgesic leads, while also acknowledging the challenges associated with their therapeutic development.
The review is well-structured, covering four key areas of μ-Conotoxin research: the mapping of interactions with NaV channel subtypes, structure-activity relationship studies, species selectivity, and the impact of disulfide connectivity on their activity. This multifaceted approach provides a thorough and nuanced understanding of the current landscape of μ-Conotoxin research, which is valuable for both researchers in the field and potential therapeutic developers.
The section focusing on the interaction of μ-Conotoxins with different NaV channel subtypes addresses the critical issue of the toxins' promiscuity and the resultant side effects, a significant barrier in their therapeutic application. The detailed discussion on structure-activity relationships further enhances our understanding of how μ-Conotoxins can be modified or optimized for better specificity and efficacy.
Additionally, the review of species selectivity and the effects of disulfide connectivity on activity are well-articulated, shedding light on some of the remarkable features of μ-Conotoxins.
In conclusion, the manuscript is a valuable contribution to the field, offering a clear, detailed, and technically sound overview of the current status of μ-Conotoxin research. It is a commendable effort to collate and analyze various facets of this research area, and it should be of interest to the scientific community engaged in NaV pharmacology and pain management research.
I have some suggestions to improve the review:
1) The section focusing on NaV selectivity (Table 1) reports The NaV subtype at which the individual peptide is the most active, but it doesn't indicate what other NaV subtypes were tested for comparison, unless they were similarly susceptible. It seems important to change this column of the table to the format Nav1.X> Nav1.Y> Nav1.Z, for these claims of selectivity to have value.
2) The section "Mapping μ-conotoxin and NaV channel interactions" needs a figure to provide structural reference for all the NaV residues.
3) line 20, "...the vast majority of cone snails are now believed to be harmless." This claim needs a reference or should be removed.
Author Response
Reviewer 2
This review delivers exactly what it promises. It adeptly addresses a crucial aspect of μ-Conotoxin research, particularly their potential as for Authors pharmacological probes and analgesic leads, while also acknowledging the challenges associated with their therapeutic development.
The review is well-structured, covering four key areas of μ-Conotoxin research: the mapping of interactions with NaV channel subtypes, structure-activity relationship studies, species selectivity, and the impact of disulfide connectivity on their activity. This multifaceted approach provides a thorough and nuanced understanding of the current landscape of μ-Conotoxin research, which is valuable for both researchers in the field and potential therapeutic developers.
The section focusing on the interaction of μ-Conotoxins with different NaV channel subtypes addresses the critical issue of the toxins' promiscuity and the resultant side effects, a significant barrier in their therapeutic application. The detailed discussion on structure-activity relationships
further enhances our understanding of how μ-Conotoxins can be modified or optimized for better specificity and efficacy.
Additionally, the review of species selectivity and the effects of disulfide connectivity on activity are well-articulated, shedding light on some of the remarkable features of μ-Conotoxins.
In conclusion, the manuscript is a valuable contribution to the field, offering a clear, detailed, and technically sound overview of the current status of μ-Conotoxin research. It is a commendable effort to collate and analyze various facets of this research area, and it should be of interest to the
scientific community engaged in NaV pharmacology and pain management research.
I have some suggestions to improve the review:
1) The section focusing on NaV selectivity (Table 1) reports The NaV subtype at which the individual peptide is the most active, but it doesn't indicate what other NaV subtypes were tested for comparison, unless they were similarly susceptible. It seems important to change this column of the
table to the format Nav1.X> Nav1.Y> Nav1.Z, for these claims of selectivity to have value.
Response: The purpose of this table is to introduce the many different μ-conotoxins and provide some background context. To comprehensively compare selectivity, it would be necessary to also list the different Nav channel subtypes, assay techniques, different μ-conotoxins as well as disulfide connectivity. As this is beyond the purpose of this table no changes have been made to the table contents. To avoid confusion for the to the reader, the title has been changed from ‘Sequences and Nav subtype selectivity of μ-conotoxins functionally characterised to date’ to ‘Summary of µ-Conotoxins functionally characterised to date’. The location of the intext reference has also been adjusted.
2) The section "Mapping μ-conotoxin and NaV channel interactions" needs a figure to provide structural reference for all the NaV residues.
Response: We thank review 2 for their suggestion. To aid readers through this section, we have updated Figure 2 to include an illustrative figure of a Nav channel and the approximate location of µ-conotoxin binding. This figure also includes some visual representation of the terminology (e.g. outer vestibule, selectivity filter etc.) used throughout this section.
We feel the level of detail require to provide graphical reference for each of the nine individual channel subtype and µ-conotoxin species would overtly complex and beyond the scope of this review.
3) line 20, "...the vast majority of cone snails are now believed to be harmless." This claim needs a reference or should be removed.
Response: Line changed to “the vast majority of cone snails are believed not to be lethal.”
Round 2
Reviewer 1 Report
Comments and Suggestions for Authors
The authors have essentially solved the questions that I referred from the previous version. It can be accepted for publication.
Author Response
We thank the reviewer for accepting our revised manuscript.
Reviewer 2 Report
Comments and Suggestions for Authors
The changes made in response to suggestions 1 and 2 are adequate. However, the response to 3 is inadequate:
3) line 20, "...the vast majority of cone snails are now believed to be harmless." This claim needs a reference or should be removed.
Response: Line changed to “the vast majority of cone snails are believed not to be lethal.”
The conjecture that the vast majority cone snails are not lethal is given no basis here. Please either support this with evidence or remove the statement. If this conjecture is wrong, people who believe it could potentially be killed by cone snails!
Author Response
We have removed the sentence "...the vast majority of cone snails are now believed to be harmless." as per the reviewer's request.